# Successive remodeling of IgG glycans using a solid-phase enzymatic platform

Yen-Pang Hsu [1,4], Deeptak Verma [2], Shuwen Sun[1], Caroline McGregor[3], Ian Mangion[1] & Benjamin F. Mann [1✉]

The success of glycoprotein-based drugs in various disease treatments has become widespread. Frequently, therapeutic glycoproteins exhibit a heterogeneous array of glycans that are intended to mimic human glycopatterns. While immunogenic responses to biologic drugs are uncommon, enabling exquisite control of glycosylation with minimized microheterogeneity would improve their safety, efficacy and bioavailability. Therefore, close attention has been drawn to the development of glycoengineering strategies to control the glycan structures. With the accumulation of knowledge about the glycan biosynthesis enzymes, enzymatic glycan remodeling provides a potential strategy to construct highly ordered glycans with improved efficiency and biocompatibility. In this study, we quantitatively evaluate more than 30 enzymes for glycoengineering immobilized immunoglobulin G, an impactful glycoprotein class in the pharmaceutical field. We demonstrate successive glycan remodeling in a solid-phase platform, which enabled IgG glycan harmonization into a series of complex-type N-glycoforms with high yield and efficiency while retaining native IgG binding affinity.

[1] Analytical Research and Development, Merck & Co., Inc, Rahway, NJ 07065, USA. [2] Computational and Structural Chemistry, Discovery Chemistry, Merck & Co., Inc, Rahway, NJ 07065, USA. [3] Process Research & Development, Merck & Co., Inc, Rahway, NJ 07065, USA. [4] Present address: Exploratory Science Center, Merck & Co., Inc, Cambridge, MA 02141, USA. ✉email: ben.mann@gmail.com

Protein glycosylation directly affects the physical and biochemical properties of proteins in eukaryotic systems[1]. According to glycoproteomic analyses, over 1% of the human genome encodes glycosylation-related enzymes and more than 50% of human proteins are glycosylated[2]. Glycoproteins carry structurally diverse oligosaccharides, called glycans, that are involved at the interface of protein-biomolecular interactions and thus determine protein stability, selectivity, and activity. The significance of protein glycosylation to biological systems has been exemplified by several diseases associated with various cancers and the immune system[3,4]. For example, patients with rheumatoid arthritis were found to have an increased galactosylation level in their serum immunoglobulin G (IgG), though the mechanism remains elusive[5]. Unsurprisingly, it follows that insights into the structure and function of glycans have yielded a profound impact on the development of therapeutic glycoproteins[6]. Manipulating glycan structures present an effective strategy to improve their efficacy and safety by modulating immunological responses, circulatory half-life, and effector functions[7,8]. Thus, glycoengineering represents a versatile tool and a great opportunity to create better medicines. To achieve this goal, technologies that enable the control of protein glycosylation profiles are essential.

However, tools to access the diverse array of glycan structures displayed in nature remain scarce, and methods that provide a high yield of the desired glycoforms have proven to be a still greater challenge to develop despite decades of study[9,10]. Through synthetic and chemoenzymatic approaches, various glycoforms have been accessed[11–13]. These structurally defined glycans can be installed onto glycoproteins through endoglycosidase and glycosynthase activities[14,15]. While this approach has advanced our ability to control protein glycosylation, the preparation of synthetic glycans becomes increasingly difficult as the number of saccharide units increases. As a result, the installation of synthetic glycans is not practical for many applications. On the other hand, genetic engineering has been applied for controlled glycan biosynthesis by either knocking out or introducing certain glycoengineering enzymes in the host cells[16]. This strategy enables in vivo glycan remodeling and has been demonstrated in non-human cell lines[17]. However, the optimization of this strategy has been impeded by the complexity of engineering glycosylation pathways. Also, microheterogeneity is often generated during glycan formation, which, although it is comparable to the natural phenomenon, does not provide exquisite control over the molecular structure[18].

In recent decades, our understanding of the in vitro activity of glycoengineering enzymes is growing rapidly[19–22]. Some of the enzymes can even function on intact glycoproteins, which opens a new window for glycan remodeling[23,24]. A remarkable example comes from the use of endoglycosidase S (Endo S) and its mutants to replace native IgG glycans with synthetic ones[25,26]. To further leverage the use of more glycoengineering enzymes, three primary challenges need to be addressed. First, characterization of enzyme activities on intact glycoproteins is required[21,27]. A comprehensive understanding of their activity, selectivity, and stability would allow researchers to design and execute glycan remodeling enzymatically. Second, preserving the integrity and functions of the substrates after the enzymatic reactions is critical, especially for therapeutic glycoproteins. Protocols with high biocompatibility are thus required. Third, to construct complex glycan structures, successive reactions using different enzymes are needed. These enzymes might require very different working conditions, such as pH and temperature. Therefore, one would need to repeat the buffer swapping and product purification processes between the enzymatic reactions, which is highly labor-intensive and time-consuming. Together, to address these needs, platforms that enable efficient, successive enzymatic glycan remodeling with high biocompatibility to the substrates are in great demand.

Inspired by solid-phase peptide synthesis (SPPS), herein, we introduce solid-phase glycan remodeling (SPGR) where enzymatic reactions are carried out on the substrates immobilized on resins[28]. This approach enables efficient reaction swapping, substrate purification, and the recovery of both products and engineering enzymes. We use human IgG as the substrate in this study because it is a major class of glycoproteins that have been applied in therapeutic development[6,29]. We quantitatively examined more than 30 glycan engineering enzymes for their activities on intact IgG immobilized on resins and then applied them in SPGR. This method has allowed us to harmonize IgG glycans into ten different glycoforms, including noncanonical structures, in 48 h with an average conversion ratio of over 95%. Physical and biochemical analyses indicated that the SPGR-engineered IgGs preserved integrity and functionality, suggesting that SPGR has high biocompatibility to the substrates.

## Results

### The design of solid-phase glycan remodeling (SPGR) for IgG glycoengineering.

Our strategy to achieve efficient, successive glycan remodeling is immobilizing IgG onto protein A resins and then executing enzymatic reactions heterogeneously (Fig. 1a, Supplementary Note 1). This enables product purification by filtration, greatly speeding up multi-step reactions. We use empty SPE (solid-phase extraction) columns with standard Luer fittings as the SPGR reaction vessels. The Luer fittings can be connected to either syringes or vacuum manifolds to control the flow speed during washing processes. A frit is inserted into the bottom of the column for trapping the solid supports. SPGR processes include 5 steps: 1) resin loading; 2) IgG immobilization; 3) washing and conditioning; 4) enzymatic glycan remodeling, and; 5) washing and elution. The third and fourth steps are repeated to swap the reaction buffer and enzymes when running multi-step glycan remodeling. The substrate IgG remains immobilized on protein A resin until all the glycan remodeling steps are complete. To characterize the glycoforms of the product IgGs, PNGase F was used to isolate the IgG glycan, followed by fluorophore labeling, solid-phase extraction (SPE) purification, and then LC-MS analyses.

To identify capable glycoengineering enzymes for SPGR, we quantitatively analyzed the activity of 34 candidates, including exoglycosidases, endoglycosidases, and glycosyltransferases (Fig. 1b, Table S1). Each enzyme was incubated with immobilized IgG for 1 or 24 h. The enzyme activity——indicated by the consumption of substrate glycan species——was then quantified via chromatographic analysis. The candidates with the highest activity in each enzyme class were selected for SPGR applications and their working conditions were further optimized (Table 1, Table S2, Fig. S1-11). Also, we defined $CR_{50}$ to be the substrate-to-enzyme molar ratio (in 1:XX format) that leads to 50% substrate conversion into the products in one hour using SPGR. This value allows us to estimate how much enzyme is required for SPGR reactions when the amount of substrate varies. It also provides the information about the reaction efficiency between different enzymes: the smaller the $CR_{50}$ value, the more efficient the reaction is.

### Trimming IgG glycans with glycosidases.

IgGs have two highly conserved glycosylation sites on the crystallizable region (Fc) at Asn 297 where more than 20 complex-type glycoforms have been found with the majority in bi-antennary structures (Fig. 1c)[30]. IgG glycans play essential roles in Fc receptor-mediated activities, such as antibody-dependent cellular cytotoxicity (ADCC)[31].

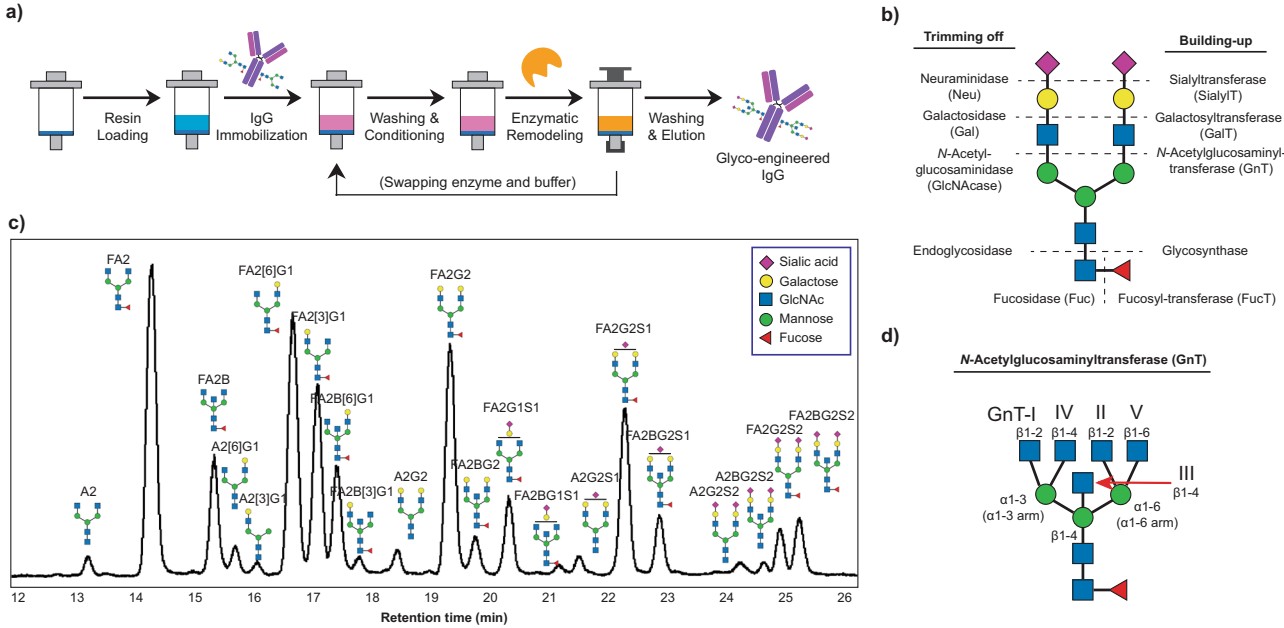

**Fig. 1 Enabling enzymatic glycoengineering using a solid-phase platform. a** Scheme of solid-phase glycan remodeling (SPGR) protocols. **b** Enzyme map for bi-antennary N-glycan glycoengineering. **c** Chromatogram of glycans collected from human serum IgG. The Oxford notation is used for glycan nomenclature. **d** The functions of *N*-acetylglucosaminyltransferases (GnTs).

**Table 1 Glycoengineering enzymes selected for SPGR and their optimized working conditions.**

| Enzyme | Source (Gene) | Function (on IgG glycans) | pH | T (°C) | Cation Co-factor | CR$_{50}$ (1:XX) |
|---|---|---|---|---|---|---|
| Neuraminidase (Sialidase) | *C. perfringens* (*nanH*) | Removes terminal α2,6-linked sialic acid | 5.5† | 42 | Ca$^{2+}$† | 0.01 |
| Galactosidase S | *S. pneumoniae* (*bgaA*) | Removes terminal β1,4-linked galactose | 5.5† | 37 | Ca$^{2+}$† | 0.047 |
| *N*-Acetylglucosaminidase S (GlcNAcase) | *S. pneumoniae* (*strH*) | Removes terminal β1,2- and β1,4-linked GlcNAc | 5.5† | 42 | Ca$^{2+}$† | 0.004 |
| Fucosidase O* | *C. omnitrophica*‡ | Removes terminal α1,6-linked fucose | 4.5† | 50† | None† | N.D. |
| Endoglycosidase S | *S. pyogenes* (*ndoS*) | Liberates glycan moiety through β1-4 GlcNAc-GlcNAc bond cleavage | 5.5† | 42 | Ca$^{2+}$† | 0.007 |
| α2-6 Sialyltransferase | *H. sapiens* (*ST6GAL1*) | Adds sialic acid to terminal galactose through α1,6 linkage | 7.5 | 37 | Mg$^{2+}$ | 0.152 |
| β1-4 Galactosyltransferase 1 | *H. sapiens* (*B4GALT1*) | Adds galactose to terminal GlcNAc through β1,4 linkage | 7.0 | 50 | Mn$^{2+}$ | 0.07 |
| *N*-Acetylglucosaminyltransferase 1* | *H. sapiens* (*MGAT1*) | Adds GlcNAc to terminal α1,3-linked mannose through β1,2 linkage | 7.5 | 30 | Na$^+$ Ca$^{2+}$ Mg$^{2+}$ Mn$^{2+}$ | 0.038 |
| *N*-Acetylglucosaminyltransferase 3 | *H. sapiens* (*MGAT3*) | Adds GlcNAc to terminal β1,4-linked mannose through β1,4 linkage | 6.5 | 30 | Na$^+$ Mn$^{2+}$ | 0.079 |
| Fucosyltransferase* | *H. sapiens* (*FUT8*) | Adds fucose to the innermost GlcNAc through α1,6 linkage | 7.0† | 37† | Na$^+$† | N.D. |

CR$_{50}$: the substrate-to-enzyme molar ratio (in XX:XX format) required for reaching 50% substrate conversion into the products in one hour using SPGR and the optimized conditions. Please refer to Table S2 for detailed working conditions. N.D.= no data. † The conditions suggested by the enzyme suppliers were used without further optimization. * Glycoengineered IgG was used as the substrate for the characterization. ‡ Genbank KXK31601.1.

About 20% of IgG glycans contain terminal sialic acids through α2-6 linkages. These sialylated glycans have been known to confer anti-inflammatory activity[32]. Similarly, IgG galactosylation modulates inflammatory properties and about 70% of the IgG glycans contain terminal β1-4 galactoses[3,5]. The galactosylation level is also known to influence the clearance rate of glycoproteins in serum, mediated by asialoglycoprotein receptors, resulting in a direct impact on their pharmacokinetic properties[33,34]. Compared to sialic acid and galactose, our understanding of *N*-Acetylglucosamine (GlcNAc)'s impacts on IgG is more limited. GlcNAc exists in all N-glycans and plays decisive roles in glycan biosynthesis pathways (Fig. 1d). Extended from the chitobiose core, GlcNAc glycosidic linkage serves as a watershed that determines the subclasses of N-glycans: complex-type, high-mannose, and hybrid-type N-glycans. Complex-type glycans can further branch into bisecting, bi-antennary, tri-antennary, and tetra-antennary glycans, and so on[35]. About 90% of human IgG glycans are bi-antennary while the rest of them have bisecting structures[30].

To control IgG glycoforms, we first aimed to harmonize them into the core saccharides by removing terminal sialic acid, galactose, and then GlcNAc (also see Supplementary Note 2.)

Neuraminidase (Neu, or sialidase) is a class of enzyme that cleaves the glycosidic linkages of sialic acids. Our screening showed that Neu from *Clostridium perfringens* has the highest activity on immobilized IgG with a $CR_{50}$ of 1:0.01. This enzyme has a broad substrate spectrum and can function on all the IgG glycoforms containing terminal sialic acid (Fig. 2, Fig. S1). Next, galactosidase (Gal) from *Streptococcus pneumoniae* showed the highest activity in our screening ($CR_{50} = 1:0.047$) for removing galactose (Fig. S2). It functions on all the IgG glycoforms containing terminal galactoses with an optimal temperature at 37 °C. To trim off GlcNAc, *N*-Acetylglucosaminidase (GlcNAcase) from *S. pneumoniae* showed the highest activity ($CR_{50} = 1:0.004$, Fig. S3). It has low glycosidic linkage selectivity and can trim terminal GlcNAc extended from the chitobiose core. Sequential treatments using these three enzymes leads to IgG glycan harmonization into (F)M3 structures (Fig. 3a).

Fucose on IgG glycan chitobiose core has been known to modulate IgG binding affinity to Fc receptors[36]. Defucosylated IgG has been reported to have an over 50-fold increase in ADCC activity[37]. As such a strong regulator, controlling the level of IgG core fucose has become an attractive strategy for improving the efficacy of IgG-based drugs. Over 90% of the human serum IgG glycans are fucosylated[30]. To identify the enzymes that can trim fucose from intact IgGs in their native confirmations, we tested seven fucosidases. Unfortunately, none of them showed an acceptable activity (Table S1). Huang et al. has reported that fucosidases only function on intact IgG when IgG glycans are trimmed down to the GlcNAc-fucose disaccharides, which indicates a strong steric interference between the enzyme and the glycan substrates[25]. Inspired by their works, we tested the fucosidase panel with glycoengineered IgG bearing (F)M3 glycans. The enzyme from *Candidatus omnitrophica* showed significantly improved activity on this group of substrates (Fig. S4, Supplementary Note 3). A 20% conversion was achieved in a 3-day reaction. The conversion ratio was further increased to 65% if non-immobilized substrates were used.

**Building IgG glycans with glycosyltransferases**. Glycosyltransferases catalyze the transfer of saccharide(s) from activated sugar phosphates, the glycosyl donors, to glycosyl acceptor molecules, such as glycoproteins[38]. Sialyltransferase (SialylT) from *Homo sapiens* exhibited the highest activity in our screening for installing sialic acid through α2-6 linkage to the IgG with terminal galactose. This enzyme has a $CR_{50}$ of 1:0.152 and apparent substrate selectivity, as shown in Fig. 2 and Fig. S5. Di-galactosylated glycan (FA2G2) and mono-galactosylated glycan with galactose at the α1-3 arm (FA2[3]G1) were completely transformed after a 16-hours reaction; while mono-galactosylated glycans at the α1-6 arm (FA2[6]G1) showed only minimal sialylation. The selective sialylation observed here agreed with previous reports and was likely caused by the folded conformation that the Fc region adopts when the galactose on the α1-6 arm is present[39,40]. Besides, we also observed a decreased enzyme activity when the (F)A2G2 glycans were mono-sialylated (Fig. S5).

To install galactose on IgG glycans, we selected the galactosyltransferase (GalT) from *Homo sapiens* (Fig. 2, Fig. S6)[41]. This enzyme catalyzed the transfer of galactose from Uridine 5'-diphosphogalactose (UDP-Gal) to IgG glycans with terminal GlcNAc. It has a $CR_{50}$ of 1:0.07 and a broad spectrum of substrate specificity that enables the transformation of all the non- and mono-galactosylated IgG glycans into bi- galactosylated forms.

The addition of GlcNAc to the chitobiose core is relatively complicated because this process involves a series of *N*-Acetylglucosaminyltransferases (GnT) with various substrate specificities (Fig. 1d)[42,43]. We investigated the activity of five human GnTs that are responsible for complex N-glycan synthesis and obtained positive results from GnT-I, III, and V. GnT-I (MGAT1) initiates the formation of the complex-type and hybrid N-linked glycans by installing an β1-2 GlcNAc to the α1-3 mannose (Fig. S7)[44]. We observed a good $CR_{50}$ of 1:0.038 in our activity screening, calculated based on (F)M3 glycan consumption. This enzyme likely possesses glycosidase activity as well because the conversion ratio reached a plateau of ~85% (Fig. S8) in all the conditions we tested. Whether or not the glycosidase activity can be repressed through genetic engineering to reach full conversion remains to be investigated. Human GnT-III (MGAT3) serves to install the bisecting GlcNAc to the β1-4 mannose through a β1-4 linkage[45]. A higher level of bisecting GlcNAc on IgG results in enhanced ADCC activity and immune cells effector functions[46]. Reactions using serum IgG as the substrate suggested that human GnT-III can function on IgG glycoforms containing at least one terminal GlcNAc (Fig. 2). However, it showed much higher activity on glycans with two terminal GlcNAc residues (Fig. S9).

Tri- and tetra-antennary N-glycans are not typically reported on native human serum IgG, and were not observed in our studies. Human GnT-V (MGAT5) is reported to add the secondary GlcNAc to the α1-6 mannose through β1-6 linkage and leads to the formation of tri-/tetra-antennary glycoforms[47]. We observed GnT-V activity after a 24-hours reaction with intact IgG, revealed by the formation of tri-antennary species. (Fig. S10). The activity of GnT-V on intact IgG is low but potentially can be improved through genetic control or evolution for future applications.

Finally, mammalian α1,6-fucosyltransferase (FucT) catalyzes the transfer of a fucose residue from GDP-fucose, the donor substrate, to the reducing-end terminal GlcNAc residue through an α1,6-linkage[48,49]. The activity of FucT on intact IgG was observed but the conversion ratio was low, likely due to the strong steric hindrance on the substrate. We further boosted the FucT reaction by increasing both enzyme concentration and incubation time. The results revealed high substrate selectivity of the enzyme, as indicated by the full consumption of A2 and A2G1 glycoforms (Fig. S11). Together, with controlled glycoforms as the substrate, modulating the level of core fucose using FucT and fucosidases on intact IgG is feasible.

**Reconstructing a harmonized glycosylation profile**. It was described how a Neu reaction followed by Gal treatment turned IgG glycans into GlcNAc-terminating glycoforms (Fig. 3a). We can further run a GlcNAcase reaction to trim the glycans into the core structures terminating with mannose. The whole sequence was performed in 24 h without isolating IgG between steps. Similarly, remodeling using Neu and then GalT generated glycans with terminal galactose (Fig. S12); while a GalT reaction followed by SialylT resulted in mono- and bi-sialylated species (Fig. S12). The demonstrated SPGR routes in this work are summarized in Fig. 4.

IgG bearing (F)M3 glycans can also serve as starting materials for rebuilding non-canonical glycoforms. Starting with the (F)M3 glycans, we applied GnT-I, GalT, and then SialylT reactions to construct a series of mono-antennary species (Fig. 3b). Mono-antennary glycans are rare in nature and their effects on protein biology remain elusive. Whether they can be utilized for regulating the interactions between IgG and Fc receptors is a great research topic of interest. In addition to mono-antennary species, we also thought to increase the population of bisecting glycans on IgG. We trimmed off terminal sialic acids and galactose from IgG and then introduced GnT-III to the resulting (F)A2 glycans. After overnight incubation at 5 μM, we reached a full conversion of IgG glycans into the bisecting forms (Fig. 3c).

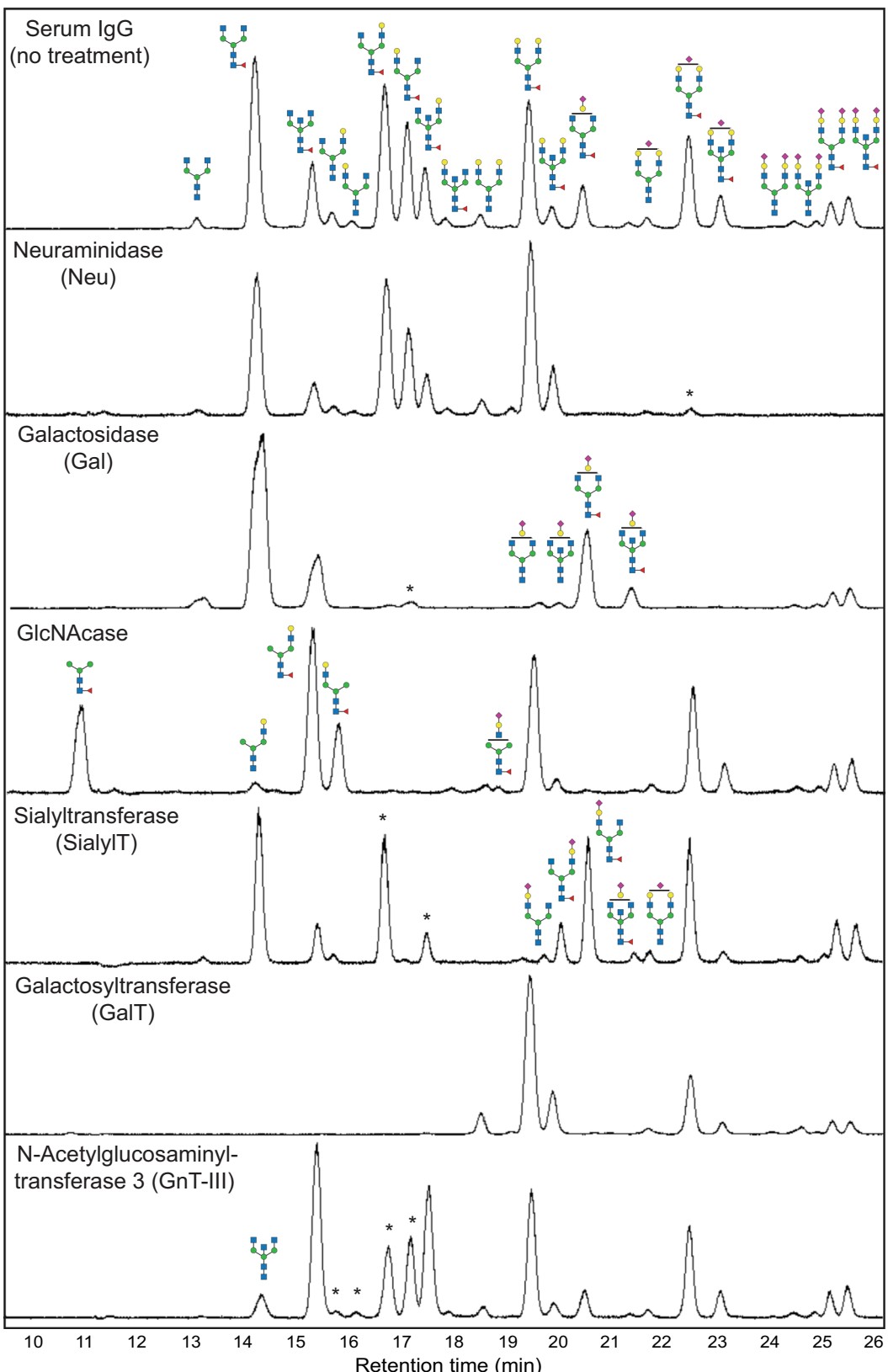

**Fig. 2 Chromatogram of glycans collected from IgG treated with different glycoengineering enzymes.** The data was collected from reactions that reached, or were close to, the plateau of the conversion. The formation of glycoforms was confirmed by mass spectrometry analyses. Please refer to Table S2 for detailed reaction conditions. Star marks indicated the substrate glycan species that have not been fully transformed in the reaction.

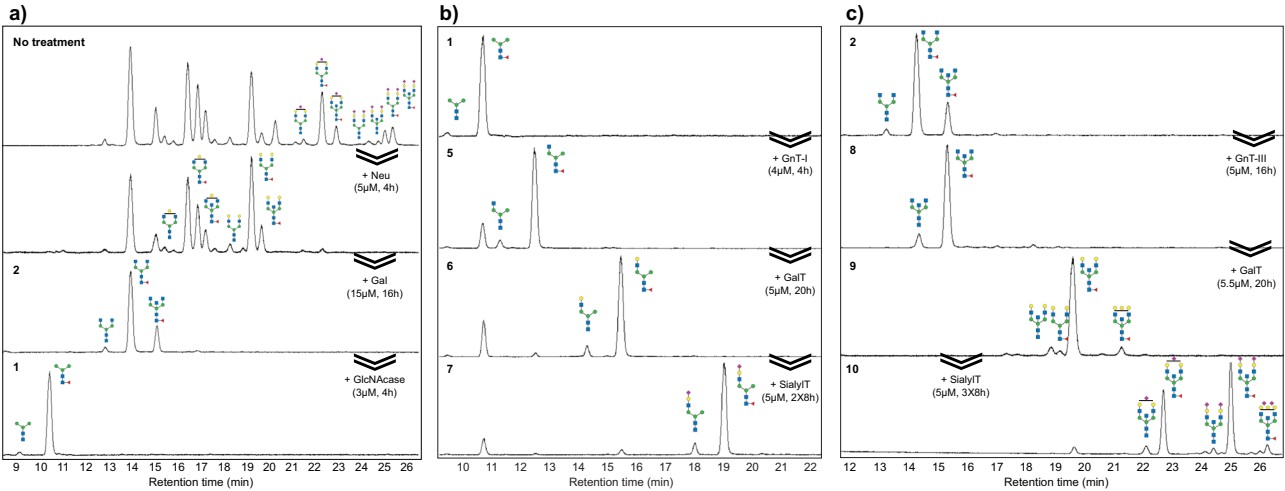

**Fig. 3 Glycan chromatograms from IgGs after sequential glycoengineering using SPGR. a** Process of remodeling IgG glycans into core saccharides (FM3 and M3 glycoforms). **b** Process of re-building core saccharides into mono-antennary species. About 10% (F)M3 glycans remained in the products due to the reversible activity of GnT-I. **c** Process of re-building FA2 and A2 glycans into bisecting species. Refer to Fig. 4 for the sample numbering and Table 1 for the buffer conditions.

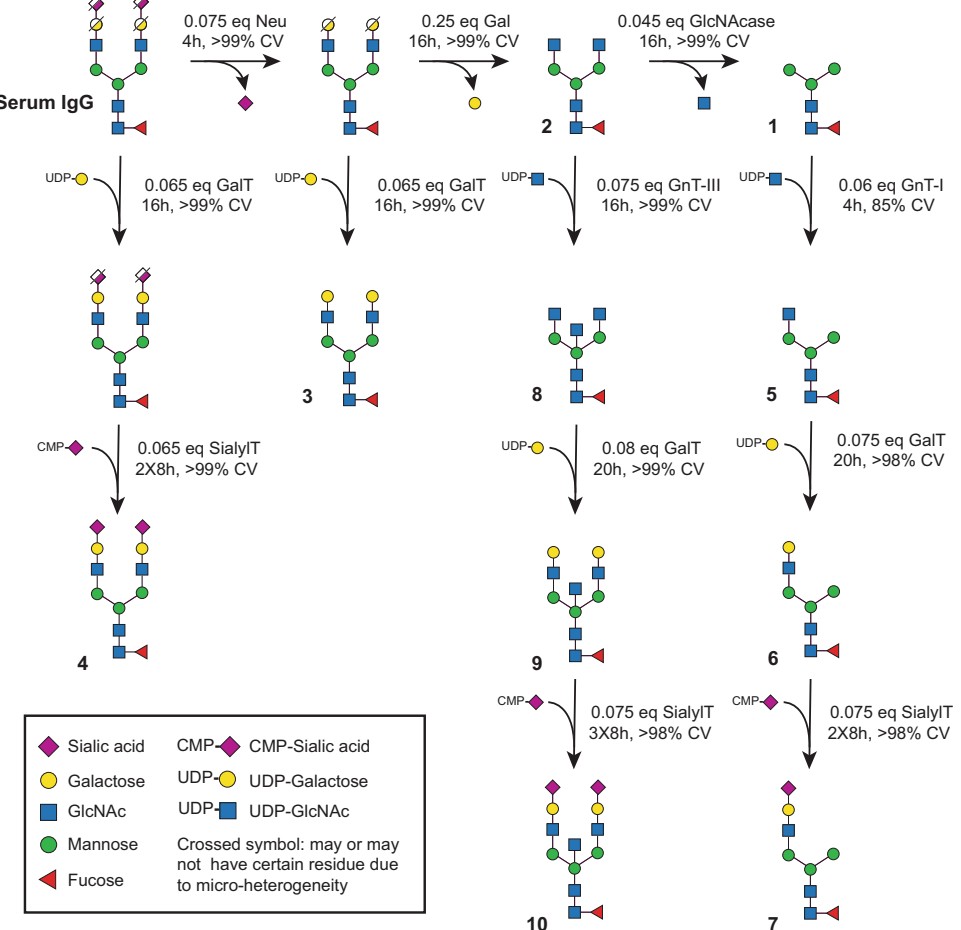

**Fig. 4 Scheme of SPGR routes investigated in this work.** Human serum IgG contains a ~5% defucosylated population and a ~10% bisecting population. For simplicity, these two populations are not shown in the scheme. The conversion (CV) ratio was calculated based on the consumption of substrate glycan species. 10 mg IgG was used to initiate the reactions with observed product recovery ranging from 75% to 80%. Reaction condition 10 mg/ml human serum IgG (66.67 μM); 1 mM sugar-donor. Neu neuraminidase, Gal galactosidase, GlcNAc N-acetyl-glucosamine, GlcNAcase N-acetylglucosaminidase, SialyIT sialyltransferase, GalT galactosyltransferase, GnT N-acetyl-glucosaminyltransferase, UDP Uridine-diphosphate, CMP cytidine-monophosphate.

These bisecting glycans can also be further galactosylated or sialylated using SPGR. Compared to mono-antennary species, both GalT and SialylT showed slightly decreased activity on bisecting glycans. Therefore, a higher enzyme concentration or longer incubation time is required to achieve full conversion.

To investigate the yield of SPGR reactions, we carefully determined the amount of the IgG substrate immobilized to protein A resin before the reactions, as well as the recovered product IgG after the reactions. Single-step reactions had a recovery ratio ranging from 78% to 85% (Fig. S13). We found that enzyme species, pH value, and buffer cation composition had no significant impact on the recovery ratio; while the use of high reaction temperature caused a slight decrease. Also, a small decrease (1–2%) of product recovery was observed with each SPGR step added. For example, 3-steps reactions for building the FM3 glycoform showed 76% recovery; and 5-steps reactions for building FA2BG2S2 glycoform had 74% recovery. In comparison to the SPGR method, a 5-steps solution-phase reaction with protein A purification after each step gave us 38.1% product recovery, which equals to 82.4% recovery in each step on average. Together, our data suggested that SPGR gives a consistent yield (product recovery) at about 80%. It presents an efficient strategy to transform IgG glycans into several mono/bi-antennary glycoforms and bisecting glycoforms.

**SPGR is biocompatible**. To examine the biocompatibility of SPGR to the substrates, we analyzed the physical properties of the glycoengineered IgGs prepared above. Size analyses using dynamic light scattering (DLS) showed no significant difference between native serum IgG and SPGR-engineered IgGs, suggesting that there was no denaturing and/or aggregation occurred during the remodeling processes (Fig. 5a). On the other hand, slight changes in their melting temperature ($T_m$) and aggregation temperature ($T_{agg}$) were observed (Fig. 5b, c, Table S3). We reasoned these changes were attributed to the altered structure of IgG. It has been known that the Asn297 glycans play roles in maintaining the conformation and stability of the Fc region through intra-molecular interactions[50,51]. Decreased $T_m$ and $T_{agg}$ values were observed in IgGs terminating with mannose (**1**) or GlcNAc (**2,5,8**), the glycoforms showing reduced intra-molecular interactions[50]. Similarly, our mono-antennary glycans (**5-7**) had lower $T_m$ values compared to others. These glycoforms lack the α1-6 arm which is important for forming intra-molecular interaction within the Fc region[39]. As expected, removing the bulk glycan structures using Endo S led to a dramatic drop in $T_m$ and $T_{agg}$.

To further exam the integrity of SPGR-engineered IgGs, we measured their binding ability to Fc receptors using a competition assay. The FRET-based assay (Förster resonance energy transfer) comprises FRET-donor and acceptor beads that are coated with unmodified IgG and Fc gamma receptor I (FcγR-I, or CD64), respectively, which results in FRET signal upon IgG-FcγR-I binding. When SPGR-engineering IgGs are introduced to the reaction, if they preserve FcγR-I binding ability, the FRET interaction is interrupted and thus gives a reduced signal. We observed a signal reduction of 60% in all the glycoengineered IgGs at the concentration of 0.1 µg/ml; while complete inhibitions were reached at about 10 µg/ml (Fig. 5d, Fig. S14). The result suggested that SPGR does not disrupt IgG integrity and its binding ability to Fc receptors. In addition, IgGs with bisecting glycoforms (**8-10**) showed decreased $EC_{50}$ values in this assay, indicating an enhanced binding affinity to FcγR-I (Table S3). Previous studies have reported that an increased level of bisecting glycoforms in mouse IgG1 results in enhanced ADCC activity, possibly owing to improved FcγR-III binding[46]. How the

bisecting GlcNAc enhances FcγR-I and III bindings, and whether or not through the same mechanisms, remain elusive but intriguing. Such a glycan-mediated regulatory effect could provide a useful handle to fine-tune the immunogenicity of IgG-based drugs through glycoengineering.

## Discussions

For decades there has been a clear demand for glycoengineering tools that enable scholars to explore the role(s) of glycan structures on the function and form of glycoconjugates. SPGR presents a straightforward strategy for controlling glycan structures with several advantages: 1) it is a biocompatible approach with minimal disruption to the protein substrates; 2) it circumvents the need to prepare synthetic glycans, which can be cumbersome; 3) tight control of glycoforms is achievable with the use of different enzyme combinations and; 4) the procedures are user-friendly and can be readily automated, greatly reducing the cost for many applications. Moreover, the idea of executing sequential enzymatic remodeling on immobilized proteins can potentially be extended to most existing biocatalytic cascade reactions involving different classes of enzymes and substrates[52].

The applicability of SPGR is largely determined by the diversity of glycoforms that we can build as well as the efficiency of the enzymatic reactions. In this work, we have reported several enzymes that have outstanding performance in SPGR reactions. However, there are still reactions of interest having low efficiency, such as fucosylation reactions. This could be addressed by the identification of enzymes with higher activity, for example, by screening more glycoengineering enzymes that have not been well-characterized, or by evolving the current SPGR enzymes toward higher reaction activity through protein engineering technologies[53,54]. In addition, there remains an opportunity to explore alternative immobilization strategies in future experiments. IgG immobilization using protein A, a 47kD protein, likely limited the enzyme efficiency by creating a strong steric hindrance. (Supplementary Note 4) Methods using oligopeptides, such as a his-tag, presumably have a lower steric effect and can enable higher enzyme activities. Since immobilization has been commonly employed for protein purification in pharmaceutical manufacturing processes, SPGR can conceivably be inserted into modern protein production processes as a "glycan modification module" to provide pure, glycoengineered proteins.

A particularly disruptive application of SPGR would be to humanize glycoforms on therapeutic proteins produced from non-human cell lines. Chinese hamster ovary (CHO) is commonly used for therapeutic protein production because they generate human-like post-translational modifications[55]. However, nonhuman glycoforms still exist in the cell line and should be removed to reduce the potential immune response in patients[56]. SPGR can be employed to humanize those glycans during the production process. From a different perspective, protein production in mammalian hosts is costly because of its long fermentation time and liability of virus infections. To address this issue, yeast has been employed as an alternative host for the large-scale expression of therapeutic proteins. Glycoproteins expressed from yeast contain high-mannose N-glycans which confer a short half-life in vivo and thereby compromise the efficacy of most therapeutic proteins[7]. Therefore, gene engineered strains are constructed for producing human-mimicking glycan patterns[57]. To date, a couple of simple glycoforms have been achieved in yeast and they have provided the desired scaffold (e.g. M5 glycan) for downstream glycan remodeling in vitro[17]. With the combination of SPGR, large-scale production of therapeutic proteins in yeast with controlled glycan structures is possible.

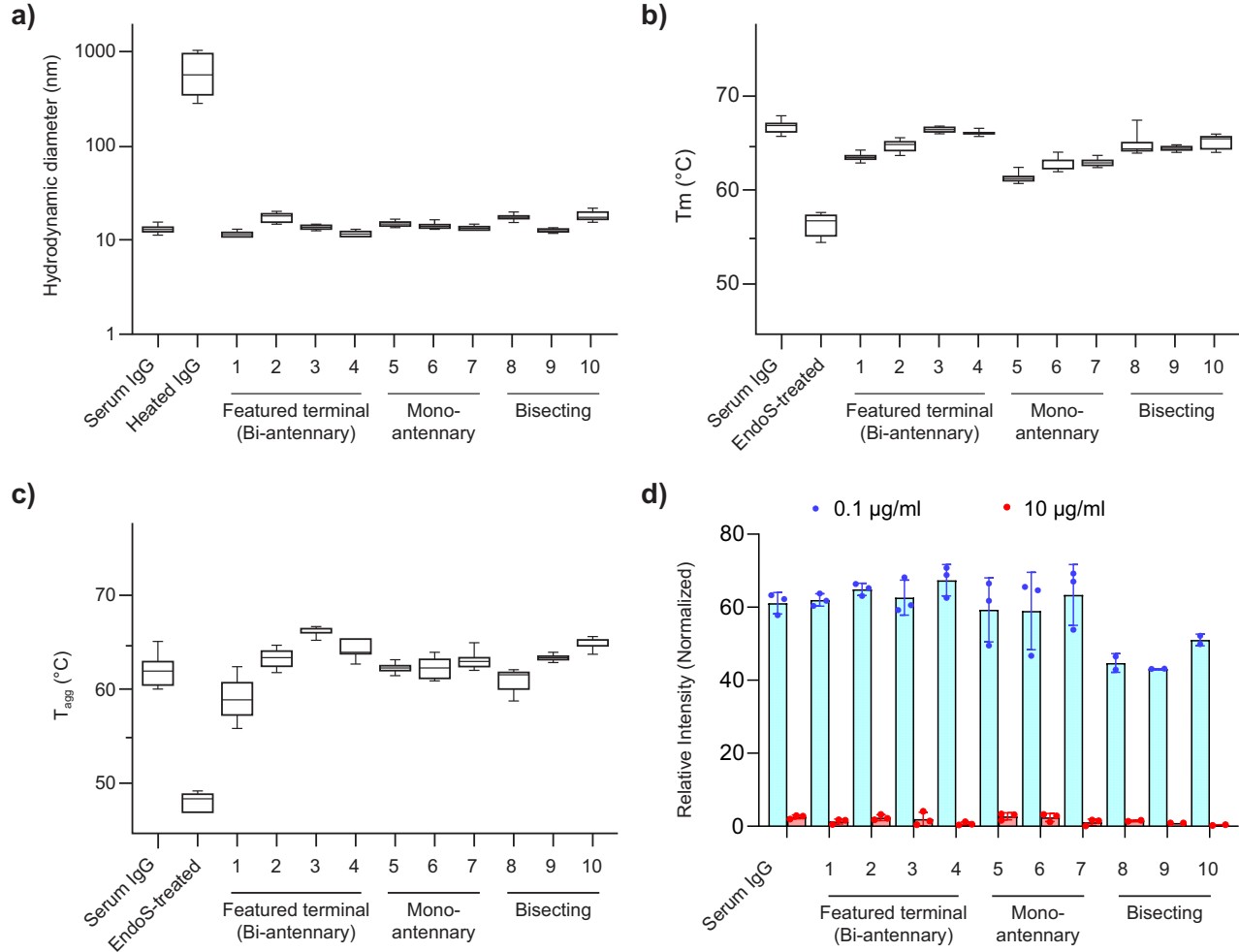

**Fig. 5 Physical and biochemical characterization of SPGR-engineered IgGs. a** Size analyses using dynamic light scattering (DLS). **b** Melting temperature analyses ($T_m$). **c** Aggregation temperature analyses ($T_{agg}$). Endo S treatment removed most IgG glycan structures, leaving only GlcNAc-Fucose di-saccharide on IgG. Box and whiskers graph: median with 25th to 75th percentile. whiskers indicate min and max values ($N = 5$). **d** Competition assay revealed the binding affinity of SPGR-engineered IgG to Fc gamma receptor I (FcγR-I). The lower the relative intensity, the stronger the interaction between SPGR-engineered IgG and FcγR-I was. Refer to Fig. 3 and S12 for the chromatograms of analyzed samples.

The awareness of public health has been raised significantly in the past months, intensifying the demand for developing better biologic medicine. Glycoproteins have proven their unignorable values in therapeutic and vaccine development, which stresses the urgent need for exquisite control of glycosylation profile for improved safety and efficacy. SPGR presents an efficient, user-friendly method for glycoengineering. It enables the control of glycan structures with various glycoforms and, presumably, on diverse glycoproteins. We believe that SPGR will expand our strategies for glycoengineering and greatly accelerate protein glycosylation studies as well as their pharmaceutical applications.

## Methods

**Solid-phase glycan remodeling (SPGR).** **(I) Resin loading:** Empty SPE columns composed of an empty column body, a frit with 0.2 μm pores, and a lid were used as reaction vessels for SPGR reactions. The columns were mounted onto a 20-wells SPE vacuum manifold. 100 μl of protein A resin (wet resin) was transferred into each column, followed by conditioning with 0.8 ml protein A-IgG binding buffer twice. A vacuum system was connected to the manifold to control the flow rate. **(II) IgG immobilization:** 1 mg (unless other specified) human serum IgG was added to 0.5 ml protein A-IgG binding buffer, followed by gently shaking until all the powder was dissolved. The solution was then transferred to the SPE column containing protein A resins. To ensure good immobilization, the columns were dismounted from the manifold, capped with Luer fittings, and then incubated for 15 min at room temperature with gentle rotating. **(III) Washing and conditioning:**

After the incubation, the columns were mounted to the manifold again, followed by washing with 0.8 ml protein A-IgG binding buffer (3 times) and then enzyme reaction buffer (2 times). After the conditioning step, the buffer was completely drained out from the columns. **(IV) Enzyme reactions:** Enzyme reaction solutions were prepared by mixing the desired amount of enzyme, reaction buffer, and saccharide donors (1 mM, for glycosyltransferase) to a final volume of 100 μl unless other specified. Please refer to Table 1 and Table S2 for the details of reaction conditions, including pH, concentration, and cation cofactors. Enzyme reactions were initiated by transferring the reaction solution to the SPE columns that contain immobilized IgG substrates. The columns were capped by Luer fitting, sealed with parafilm, and incubated at temperature-controlled shakers for a certain amount of time (Table S2). After the reaction, the enzyme solution was discarded (or recovered) and the columns were washed 6–8 times with 0.8 ml protein A-IgG binding buffer via the extraction manifold. The (III) and (IV) steps were repeated until all the glycan remodeling steps were done. **(V) Elution of glycoengineered IgG:** 250 μl protein A-IgG elution buffer was added to the reaction columns and incubated for 2 minutes at room temperature. After collecting the eluent, another 250 μl elution buffer was added to the column and the process was repeated. The two portions of collected eluent were combined into a 0.5 ml MWCO (molecular weight cut-off) tube and concentrated via centrifugation at 14,000 g for 5 minutes, followed by buffer exchange into 50 mM HEPES buffer (pH 8). After adjusting the concentration of the eluted IgG substrate to 3 mg/ml using NanoDrop, the IgG substrates were stored at 4 °C and were ready for analysis. IgG concentration was adjusted in this step in order to ensure the same amount of sample was charged to the downstream analyses. The IgG recovery ratio of each glycoengineering reaction using this method is approximately 80% (the yields could vary when different resin, buffer, or substrate IgG are used.) For multi-step SPGR reactions shown in Fig. 4, 10 mg human IgG was used to initiate the reactions with a final material recovery ranging from 75 to 80%.

**Table 2 Methods for glycan chromatograpgy in this study.**

**Mobile phase A: 50 mM ammonium formate in H$_2$O, pH 4.4**

**Mobile phase B: 100% Acetonitrile**

**Temperature: 60 °C**

**Injection volume: 5–10 µl**

| Time (min) | Flow rate (ml/min) | %A | %B |
|---|---|---|---|
| 0 | 0.4 | 25 | 75 |
| 35 | 0.4 | 46 | 54 |
| 36.5 | 0.2 | 100 | 0 |
| 39.5 | 0.2 | 100 | 0 |
| 43.1 | 0.2 | 25 | 75 |
| 47.6 | 0.4 | 25 | 75 |
| 55 | 0.4 | 25 | 75 |

An Agilent 1290 Infinity II LC system tandem with Agilent 6500 Series quadrupole time-of-flight MS was used for glycan analysis. ACQUITY BEH Glycan column (130 Å, 1.7 µm, 2.1 × 150 mm) was used for optimal separation of IgG glycoforms.

**LC-MS Analysis of IgG glycans**. This protocol is adapted from the Glycoworks manual provided by Waters. **(I) Glycan isolation:** 7.5 µl IgG substrate (3 mg/ml, prepared as described above), 6 µl RapiGest SF (50 mg/ml, in Glycoworks buffer), and 15.3 µl water were mixed in a 1.5 ml microtube. The mixture was then incubated at 90 °C for 5 min to denature the substrates. After the samples were cooled down to room temperature, 1.2 µl Rapid PNGase F was added to the tube, followed by another incubation at 50 °C for 10 mi. **(II) Glycan labeling:** After PNGase F digestion, 12 µl RapiFluor-MS solution (70 mg/ml, in DMF) was added to the solution. The mixture was gently vortexed and then incubated at room temperature for 20 min without any light exposure. **(III) Glycan purification:** After labeling, the samples were diluted with 360 µl acetonitrile (1:9 volume ratio). Oasis SPE µPlate from Waters (along with the use of µPlate extraction manifold) was employed for the 1st solid-phase extraction purification, and Discovery SPE from Sigma-Aldrich (along with the use of 20-wells SPE vacuum manifold) was used for the 2nd purification to ensure high signal-to-noise ratio: The SPE columns/µPlate were first washed by water (1 column volume) and then conditioned by water-acetonitrile solution (10:90 v/v, 1 column volume). The glycan samples (in 90% acetonitrile solution) were then charged to the column/µPlate, followed by washing with washing buffer (formic acid/water/acetonitrile 1:9:90 v/v/v, 2 column volume). The glycans were then eluted using 60 µl elution buffer (200 mM ammonium acetate in 5% acetonitrile). **(IV) HILIC-MS analysis:** The purified glycan samples were injected to UPLC equipped with ACQUITY BEH Glycan column (130 Å, 1.7 µm, 2.1 × 150 mm) tandem with IMQ-TOF MS for glycan profile analysis. Please refer to the literature published by Pucic et al. and Kristic et al. for detailed peak assignment[30,58]. The method provided by Waters for glycan chromatography was used in this work (Table 2).

**Conversion ratio quantification**. The conversion ratio of each SPGR reaction was calculated based on the consumption of the substrate glycan species. UV absorption at 260 nm (RapiFluor-MS) from chromatography analysis was used for the quantification of glycan populations. We first normalized the chromatographic peak area of the substrate glycan species to the total glycan peak area (Eq. 1). This gives us the percentage of substrate glycan population. The reduction of substrate glycan population after the reaction was then divided by the initial value to calculate the percentage of substrate conversion (Eq. 2). We assume that there is no glycan shedding off from IgG during the experiments (namely, no endoglycosidase activity). Please refer to Table S4 for detailed substrate species used in the calculation.

For endoglycosidase reactions, their activity was quantified by absolute glycan quantification using an internal standard. This is because endoglycosidases' activity results in the reduction of all glycan signals in the chromatography analyses. A known amount of internal standard (Glycan quantitative standard, Waters) was added to the sample to measure the amount of IgG glycan before and after the reactions.

$$\text{Substrate glycan population} = \frac{\sum \text{Peak area}_{\text{substrate1,2,3}\ldots i}}{\sum \text{Peak area}_{\text{all glycans}}} = R_s \quad (1)$$

$$\text{Conversion Ratio} = \frac{R_{S(\text{initial})} - R_{S(\text{final})}}{R_{s(\text{initial})}} \quad (2)$$

**CR$_{50}$ calculation**. CR$_{50}$ is defined as the substrate-to-enzyme molar ratio (in XX:XX format) that leads to 50% substrate species conversion into the products in 1 h at optimized working conditions (temperature, pH, cation) using SPGR. This value is determined based on dose-dependent experiments where the conversion ratio at different enzyme concentrations in 1 hour was tested. A sigmoidal curve fitting ([agonist] vs normalized response, GraphPad Prism, see the equation below) was applied to the data for calculating the enzyme concentration that gives 50% substrate conversion. The resulting enzyme concentration is divided by the substrate (IgG) concentration to give CR$_{50}$.

$$Y = \frac{100X}{CR_{50} + X}, \quad (3)$$

where Y is normalized response from 0 to 100; X is the concentration of enzyme; X is equal to CR$_{50}$ at Y = 50.

**Physical property characterization of SPGR-engineered IgGs**. Dynamic light scattering (DLS), melting temperature (T$_m$) and aggregation temperature (T$_{agg}$) studies were executed using UNcle (Unchainedlabs). Glycoengineered IgGs from SPGR reactions were eluted from protein A resins, followed by buffer exchange into HEPES buffer as described above. The protein concentration was then adjusted to 1 mg/ml using NanoDrop. 9 µl of the purified IgG samples were injected into UNcle sample holders (5 replicates for each sample). DLS measurement was performed at 25 °C (4 acquisition, 5 seconds each). Static light scattering (SLS) for T$_m$ and T$_{agg}$ measurement was carried out from 25 °C to 90 °C with a temperature increase of 0.3 °C per minute.

Please note that all the tested IgG samples (**1-10**) contained a ~5% defucosylated population. The biantennary samples (**2-4**) had ~10% bisecting glycoforms. The mono-antennary samples (**5-7**) had ~10% (F)M3 glycans due to the reversible activity of GnT-I. Sialylated samples (**4** & **10**) possessed about 1:1 mono- and di-sialylated populations. Please refer to Fig. 3 and S12 for detailed glycans species and population distributions.

**Binding assays between glycoengineered IgGs and Fc gamma receptor 1**. This protocol is adapted from the AlphaLISA human FCGR binding kit manual provided by PerkinElmer. Briefly, serial dilutions of SPGR-engineered IgGs with 1X HiBlock buffer were prepared with the highest concentration at 1 mg/ml and the lowest concentration at 0.1 µg/ml. 10 µL of each diluted IgG samples were mixed with 10 µL 4X human FcγR1 solution and 20 µL Donor/acceptor beads solution into a white 96-well plate. The plate was then sealed and incubated at 25 °C for 90 min without any light exposure. After the reaction, the fluorescence signal at 615 nm was determined using an EnVision Multimode Plate Reader (equipped with AlphaScreen module).

**Computational modeling of IgG-protein A complex**. Protein A homology model was constructed using the Swiss-Model server[59]. PDB structure 5H7B, which has 79.5% sequence identity with Protein A, was used as a template to construct the homology model[60]. A visual inspection of Protein A model illustrated 4 distinct IgG binding domains. PDB structure 5U4Y was used as a template to identify the spatial positioning of the full-length Protein A and the IgG[61]. The template structure contains only the B-domain of the protein A molecule. The spatial position of the B-domain helped us overlay the full-length protein-A molecule and allowed us to identify steric hindrances between other protein-A domains and the IgG molecule. Structure overlay and the movie illustrating clashes between the molecules was generated using Pymol (Molecular Graphics System, Version 2.0, Schrödinger LLC).

**Plotting and graphic**. Data plotting and curve fitting were done by using GraphPad Prism 8. Figures and cartoons were created by Adobe Illustrator.

**Statistics and Reproducibility**. All the experiments in this work have been reproduced at least 2 times (N = 3) unless specified in the figure legends. Consistent results were obtained. Statistical analyses were performed using GraphPad Prism 8 with all the collected data points included.

**Reporting summary**. Further information on research design is available in the Nature Research Reporting Summary linked to this article.

## Data availability

Source data underlying the graphs and charts presented in this work are available in the Supplementary Data 1. Raw data (mass spectrum, chromatogram) is available upon request to the authors.

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

## Acknowledgements

We thank the Merck Research Laboratory (MRL), Merck Postdoctoral Research Fellows Program, and Analytical Research and Development (AR&D) department for the financial support. We are also thankful to Anumita Saha (Data-Rich Measurement group), Alycia Shoultz (Protein Engineering group), Jacob Forstater (Chemistry Enabling Technology group), Fuh-Rong Tsay (Method Screening and Purification group) and Sampat Ingale (Exploratory Science Center) for technical assistance and advice.

## Author contributions

SPGR method development, enzyme activity screening/characterization, and data analyses were performed by Y.-P.H. and B.F.M. Computational modeling was done by D.V. Y.-P.H., D.V., S.S., C.M., I.M., and B.F.M. were involved in the design of the research and manuscript drafting.

## Competing interests

The authors declare no competing interests.
