## [Peer Review File · Communications Biology]

Reviewers' comments:

Reviewer #1 (Remarks to the Author):

This work describes a new approach for enzymatic glycoengineering of antibodies based on the immobilization of antibody substrates. By optimizing reaction conditions for each enzyme, a one-step direct glycoengineering for antibody was achieved. A broad panel of over 30 enzymes was evaluated in this method and some worked very well. Excellent conversion yields were obtained for galactosylation and the challenging terminal sialylation. Although chemoenzymatic glycoengineering has been well studied and developed in solution phase, the solid-phase based one-step system shows advantages in reaction feasibility and product purification. This is a facile method and could be widely used in most important glyco-remodeling of antibodies. There are several concerns that should be addressed before it can be considered for publication.

1. The enzymatic reactions in this work were slow and usually takes several days to complete, especially for sialylation. This is not an efficient way for antibody glycoengineering. What efforts did the authors tried to improve the enzyme efficiency? A discussion about potential methods for improvement would be good.

2. Conversion yield was used throughout the manuscript. It shows high conversion ratio for many enzymatic reactions in this work, however it lacks the important information about the product. This is especially true when a reaction is run for several days. After such long time period, I'm how much antibodies were actually recovered from the elution of immobilized column. This piece of important information that can't be implied from the HPLC glycan analysis of eluted antibodies.

3. What is the reaction scale for most of the enzymatic reaction in this work? It is important for an immobilization method to access large scale reactions, like 10 mg for antibody. It would be more practice and useful if the method can be performed on a large scale.

Reviewer #2 (Remarks to the Author):

In this manuscript, Mann and co-workers developed a solid-phase glycan remodeling (SPGR) platform for IgG glycan engineering. In this approach, IgGs are first immobilized on resins, then they are subjected to glycosidase-mediated glycan trimming to generate homogenous glycoepitopes, which can be modified by recombinant glycosyltransferases to add glycans back to form glycan epitope-defined IgGs for studying their biological functions. Although solid phase-based glycan arrays have been developed for 20 years already, solid phase-based glycoprotein modifications are rarely investigated. This report, therefore, represents a timely study.

However, there are a few issues the authors may need to take into consideration:

1. Figure 4 needs to be updated to make the workflow clearer. I understand that bracket represents "with or without specific residues", but it is very confusing. Seems to me that the same structures remain there after glycosyltransferase-mediated modifications.

2. Please comment on the advantages of the step-wise trimming vs. the one-step trimming using Endo-F/H directly.

3. Citation format is not consistent. Very difficult to read. In certain references, all authors are listed, but not in others. I think a few important references are missing, especially those from the Moremen, the Boons (at UGA) and the Wu labs (at Scripps). The enzymes developed by Moremen are probably the most powerful ones for glycans engineering:

<https://pubmed.ncbi.nlm.nih.gov/29251719/>.

4. I would not use Fuc as the abbreviation for fucosidases. It is usually used as the abbreviation for fucose.

Reviewer #3 (Remarks to the Author):

This manuscript is well organized and novel, it is of great interest to the glycoscience and protein/antibody modification field. For the authors, I just have several minor questions:

1. When IgG is fixed to protein A resin, will the resin affect the efficiency of different types of glycotransferase and glycosidase on IgG, and whether the immobilization efficiency of IgG to protein A is affected by the reaction condition?

2, Extensive screening works have been performed on the reaction conditions of different glycan-engineered enzymes, since the most appropriate reaction temperature and PH (5.5-7.5) value are different among various enzymes, how to ensure that the reaction conditions for the previous enzyme does not affect the reaction of the next one?.

3, For continuous reaction, the replacement of metal ion types is more cumbersome, is it possible to use one or two metal ions to simplify the glycosylation step?

Response to reviewers:

Reviewer #1 (Remarks to the Author):

This work describes a new approach for enzymatic glycoengineering of antibodies based on the immobilization of antibody substrates. By optimizing reaction conditions for each enzyme, a one-step direct glycoengineering for antibody was achieved. A broad panel of over 30 enzymes was evaluated in this method and some worked very well. Excellent conversion yields were obtained for galactosylation and the challenging terminal sialylation. Although chemoenzymatic glycoengineering has been well studied and developed in solution phase, the solid-phase based one-step system shows advantages in reaction feasibility and product purification. This is a facile method and could be widely used in most important glyco-remodeling of antibodies. There are several concerns that should be addressed before it can be considered for publication.

We thank the reviewer for his appreciation of the design and feasibility of the solid-phase platform. We aimed to develop this platform into a user-friendly approach that allows researchers from different fields, with various purposes, to easily remodel glycans and investigate the relationships between discrete glycoforms and antibody/protein behaviors. We truly believe it will accelerate the advance of glycosylation and protein biology research.

1. The enzymatic reactions in this work were slow and usually takes several days to complete, especially for sialylation. This is not an efficient way for antibody glycoengineering. What efforts did the authors tried to improve the enzyme efficiency? A discussion about potential methods for improvement would be good.

We agree with the reviewer that the applicability of the current platform is restricted by the limited activity of certain reactions, such as the fucosidase reaction and sialyltransferase reaction. To address this issue, glycoengineering enzymes with higher activity on intact IgG are required. This could be achieved by screening more glycoengineering enzymes that have not been well-characterized, or by employing protein engineering technologies to evolve current SPGR enzymes toward higher reaction activity. We have initiated internal collaborations with the Merck protein engineering group for improving the activity of fucose- and GlcNAc-related glycoengineering enzymes. We aim to develop this platform into a versatile tool that enables efficient glycan remodeling into various glycoforms for glycoprotein therapeutic development as well as fundamental protein glycosylation research. Another opportunity to improve SPGR reactions is the use of different substrate immobilization methods. Our data (refer to supplementary discussion #4) suggested that protein A (47 kD) immobilization creates steric hindrance on IgG, which compromises the activity of glycoengineering enzymes. The use of Histidine-Nickel resin immobilization might address this limitation as it creates less steric hindrance on the substrate. Thus, we believe immobilization methods are worth to be investigated in the future.

To deliver the above information to the readers, we have expanded the discussion section of the manuscript, as shown below: (reference citations included in the main text)

“The applicability of SPGR is largely determined by the diversity of glycoforms that we can build as well as the efficiency of the enzymatic reactions. In this work, we have reported several enzymes that have outstanding performance in SPGR reactions. However, there are still reactions of interest having low

efficiency, such as fucosylation reactions. This could be addressed by the identification of enzymes with higher activity, for example, by screening more glycoengineering enzymes that have not been well-characterized, or by evolving the current SPGR enzymes toward higher reaction activity through protein engineering technologies. In addition, there remains an opportunity to explore alternative immobilization strategies in future experiments. IgG immobilization using protein A, a 47kD protein, likely limited the enzyme efficiency by creating a strong steric hindrance (also see supplementary discussions). Methods using oligopeptides, such as a his-tag, presumably have a lower steric effect and can enable higher enzyme activities. Since immobilization has been commonly employed for protein purification in pharmaceutical manufacturing processes, SPGR can conceivably be inserted into modern protein production processes as a “glycan modification module” to provide pure, glycoengineered proteins.”

2. Conversion yield was used throughout the manuscript. It shows high conversion ratio for many enzymatic reactions in this work, however it lacks the important information about the product. This is especially true when a reaction is run for several days. After such long time period, I'm how much antibodies were actually recovered from the elution of immobilized column. This piece of important information that can't be implied from the HPLC glycan analysis of eluted antibodies.

We thank the reviewer for the comments. To answer this question, we performed single-step and multi-step SPGR reactions using the enzymes selected in this work and their optimized reaction conditions. We carefully determined the amount of the serum IgG substrate immobilized to the protein A resin before the reactions, as well as the recovered product IgG after the reactions (See Figure A below). Single-step reactions had a recovery ratio ranging from 78% to 85%. Glycoengineering enzymes did not impact product recovery as the control reactions without the enzymes showed consistent recovery ratio. Also, the selection of pH value and cation cofactors has no significant effects. By contrast, the recovery ratio decreased slightly when the reaction temperature and incubation time were increased, which might be attributable to the accelerated protein degradation (denaturation, precipitation, or aggregation) at elevated temperature (Figure B).

For multi-step SPGR reactions, 3-steps reactions for building the FM3 glycoform showed 76% recovery; while 5-steps reactions for building FA2BG2S2 glycoform had 74% recovery. A small decrease (1-2%) of product recovery was observed with each SPGR step added. In comparison, the same 5-steps reaction without using SPGR—instead, performing protein A purification after each step (5 times in total)—gave us 38.1% product recovery, which equals to 82.4% recovery in each step on average. Together, our data suggested that the yield (product recovery) of the SPGR reactions is largely determined by the purification protocols (e.g. number of steps). Running sequential reactions using SPGR could improve the yield of the product significantly. Also, the enzymatic reactions reported in this work showed minimal effects on product recovery.

To deliver this information to the readers, we have added the data to the ESI (Supplementary Figure 13) as well as a new paragraph in the main text, as shown below:

“To investigate the yield of SPGR reactions, we carefully determined the amount of the IgG substrate immobilized to protein A resin before the reactions, as well as the recovered product IgG after the reactions. Single-step reactions had a recovery ratio ranging from 78% to 85% (Figure S13). We found that enzyme species, pH value, and buffer cation composition had no significant impact on the recovery ratio; while the

use of high reaction temperature caused a slight decrease. Also, a small decrease (1-2%) of product recovery was observed with each SPGR step added. For example, 3-steps reactions for building the FM3 glycoform showed 76% recovery; and 5-steps reactions for building FA2BG2S2 glycoform had 74% recovery. In comparison to the SPGR method, a 5-steps solution-phase reaction with protein A purification after each step gave us 38.1% product recovery, which equals to 82.4% recovery in each step on average. Together, our data suggested that SPGR gives a consistent yield (product recovery) at about 80%. It presents an efficient strategy to transform IgG glycans into several mono/bi-antennary glycoforms and bisecting glycoforms.”

Figure Legends: Studies of material recovery after SPGR reactions. (A) Recovery ratio after single-step reactions and multi-step reactions. Optimized reaction conditions that ensure full conversion were used, as summarized in the inserted table. In control experiments, glycoengineering enzymes were not added to the reactions. Reaction scale: 1 mg IgG. (B) High reaction temperature resulted in slight decrease of material recovery. Error bars: mean and standard deviation (N=2). Abbreviations: Neu: neuraminidase; Gal: galactosidase; GlcNAcase: *N*-Acetylglucosaminidase; SialyIT: sialyltransferase; GalT: galactosyltransferase; GnT-III: *N*-Acetylglucosaminyltransferase III.

3. What is the reaction scale for most of the enzymatic reaction in this work? It is important for an immobilization method to access large scale reactions, like 10 mg for antibody. It would be more practice and useful if the method can be performed on a large scale.

Thanks for the question. We have added a brief description of the reaction scales in our experiments. For enzyme characterization and working condition optimization experiments, 1 mg of serum IgG was used, while 10 mg IgG was used to initiate the multi-step SPGR reactions shown in Figure 4. This information is now better described in both the main text and the supplementary information:

Figure 4 legends: “10 mg IgG was used to initiate the reactions with observed product recovery ranging from 75% to 80%.”

Supplementary Table 1: “Enzyme activity was determined by using SPGR protocol with 1mg human serum IgG (66.7 μ M) and glycoengineering enzyme (0.25 μ M) for 1 or 24 hours.”

Supplementary Method: “The IgG recovery rate of each glycoengineering reaction using this method is approximately 80% (the yields could vary when different resin, buffer, or substrate IgG are used.) For multi-step SPGR reactions shown in Figure 4, 10 mg human IgG was used to initiate the reactions with a final material recovery ranging from 75 to 80%”

We have yet to study SPGR at the clinical and manufacturing scale and are looking for opportunities to implement SPGR for internal or external pipeline projects as we continue to develop applications of the technique.

Reviewer #2 (Remarks to the Author):

In this manuscript, Mann and co-workers developed a solid-phase glycan remodeling (SPGR) platform for IgG glycan engineering. In this approach, IgGs are first immobilized on resins, then they are subjected to glycosidase-mediated glycan trimming to generate homogenous glycoepitopes, which can be modified by recombinant glycosyltransferases to add glycans back to form glycan epitope-defined IgGs for studying their biological functions. Although solid phase-based glycan arrays have been developed for 20 years already, solid phase-based glycoprotein modifications are rarely investigated. This report, therefore, represents a timely study.

However, there are a few issues the authors may need to take into consideration:

1. Figure 4 needs to be updated to make the workflow clearer. I understand that bracket represents “with or without specific residues”, but it is very confusing. Seems to me that the same structures remain there after glycosyltransferase-mediated modifications.

We thank the reviewer for pointing out this issue. We have revised Figure 4 to make it clearer for the readers. The changes we have made include: (also see inserted figure)

- Replaced “bracket items” with crossed symbols to better deliver the message that certain sugar residues might not exist in all the substrate molecules due to micro-heterogeneity.
- Included side arrows to indicate the sugar residues being removed from or added to the substrate.
- Changed the way we present enzyme concentration. Molarity was replaced with enzyme-to-substrate equivalents.
- Graphic legends were added to better indicate the identity of each component.

Figure Legends: Scheme of SPGR routes investigated in this work. Human serum IgG contains a ~5% defucosylated population and a ~10% bisecting population. For simplicity, these two populations are not shown in the scheme. The conversion (CV) ratio was calculated based on the consumption of substrate glycan species. 10 mg IgG was used to initiate the reactions with observed product recovery ranging from 75% to 80%. Reaction condition: 10 mg/ml human serum IgG (66.67 μ M); 1 mM sugar-donor. Abbreviations: Neu: neuraminidase; Gal: galactosidase; GlcNAc: N-acetyl-glucosamine; GlcNAcase: N-acetylglucosaminidase; SialyIT: sialyltransferase; GalT: galactosyltransferase; GnT: N-acetylglucosaminyltransferase; UDP: Uridine-diphosphate; CMP: cytidine-monophosphate.

2. Please comment on the advantages of the step-wise trimming vs. the one-step trimming using Endo-F/H directly.

Thanks for the thoughtful question. Stepwise removal of terminal residues using exo glycosidases gives ready access to N-glycan motifs for downstream SPGR modifications. In comparison, endoglycosidases specifically cleave the β 1-4 linkage between the GlcNAc residues in the chitobiose core, leading to the removal of the glycan majority in one step. They serve as powerful tools for efficiently trimming glycans to the first N-acetylglucosamine. When synthetic glycans (glycan oxazolines) are available, endoglycosidases may be employed in SPGR for chemoenzymatic engineering to replace native IgG glycans. (Huang *et al. JACS*, 2012) Advantages exist for both step-wise and one-step trimming approaches, depending on the chosen substrate and desired products. Considering human IgG as substrate, a step-wise approach to trimming and rebuilding can be used to efficiently access terminally sialylated and galactosylated structures under aqueous conditions, taking advantage of the chitobiose core and antennae structures that remain intact. However, steric hindrances may limit the utility of both glycosidase and glycosyltransferase enzymes on a protein substrate on a case-by-case basis. Thus, both multi- and single-step approaches should be used as needed. In our work, We have characterized the activity of six endoglycosidases (Table S1, Figure S16). Endo S showed the highest activity on fully folded IgG, presenting a powerful tool for SPGR applications. The discussion about endoglycosidases has been included in Supplementary Discussion #2, as shown below: (reference citations included in the supplementary material)

*“IgG glycan trimming can also be implemented through the chitobiose core GlcNAc residues. Endoglycosidases specifically cleave the β 1-4 linkage between the GlcNAc residues in the chitobiose core, shaving off the majority of glycans. This makes them powerful tools when the removal of IgG glycan is needed. They have also been employed for chemoenzymatic glycan modification where native glycans of glycoproteins are removed by endoglycosidases, followed by synthetic glycan installation back to the proteins using mutated endoglycosidases (also known as glycosynthases). To know whether endoglycosidases can be applied to SPGR, we analyzed the activity of endoglycosidases on intact IgG. Of the six tested enzymes, the candidate from *Streptococcus pyogenes* (known as Endo S) exhibited the highest conversion ratio on intact IgG (CR50=1:0.007, Figure S16). It has been known that endoglycosidases have different preferences for substrate structures. In agreement with reported studies, Endo S effectively liberates N-glycans from human IgG in SPGR. On the other hand, the ones with higher specificity to high-mannose glycans, such as endoglycosidase D, did not show detectable IgG glycan conversion in our screening (Table S1). A complete Endo S reaction led to the removal of glycan majority, giving a relatively clean chromatogram as shown in Figure S16”*

3. Citation format is not consistent. Very difficult to read. In certain references, all authors are listed, but not in others. I think a few important references are missing, especially those from the Moremen, the Boons (at UGA) and the Wu labs (at Scripps). The enzymes developed by Moremen are probably the most powerful ones for glycans engineering: <https://pubmed.ncbi.nlm.nih.gov/29251719/>.

We thank the reviewer for pointing out this issue. We have revised the reference section and have it match the *Nature* format per the requirement of *Communications Biology*. We also added a couple of

references that are strongly related to our work from the groups that the reviewer suggested, as shown below:

- 12 Wang, Z. *et al.* A general strategy for the chemoenzymatic synthesis of asymmetrically branched N-glycans. *Science* **341**, 379-383, doi:10.1126/science.1236231 (2013).
- 13 Li, T. *et al.* An automated platform for the enzyme-mediated assembly of complex oligosaccharides. *Nature Chemistry* **11**, 229-236, doi:10.1038/s41557-019-0219-8 (2019).
- 22 Liu, L. *et al.* Streamlining the chemoenzymatic synthesis of complex N-glycans by a stop and go strategy. *Nature Chemistry* **11**, 161-169, doi:10.1038/s41557-018-0188-3 (2019).
- 43 Gagarinov, I. A. *et al.* Chemoenzymatic Approach for the Preparation of Asymmetric Bi-, Tri-, and Tetra-Antennary N-Glycans from a Common Precursor. *Journal of the American Chemical Society* **139**, 1011-1018, doi:10.1021/jacs.6b12080 (2017).
- 49 Boruah, B. M. *et al.* Characterizing human α -1,6-fucosyltransferase (FUT8) substrate specificity and structural similarities with related fucosyltransferases. *J Biol Chem* **295**, 17027-17045, doi:10.1074/jbc.RA120.014625 (2020).
- 53 Moremen, K. W. & Haltiwanger, R. S. Emerging structural insights into glycosyltransferase-mediated synthesis of glycans. *Nature Chemical Biology* **15**, 853-864, doi:10.1038/s41589-019-0350-2 (2019).
- 54 Taujale, R. *et al.* Deep evolutionary analysis reveals the design principles of fold A glycosyltransferases. *eLife* **9**, e54532, doi:10.7554/eLife.54532 (2020).

4. I would not use Fuc as the abbreviation for fucosidases. It is usually used as the abbreviation for fucose.

We thank the reviewer for the suggestion. We have removed the abbreviation and replaced it with the full name of the enzyme, fucosidase.

Reviewer #3 (Remarks to the Author):

This manuscript is well organized and novel, it is of great interest to the glycoscience and protein/antibody modification field. For the authors, I just have several minor questions:

1. When IgG is fixed to protein A resin, will the resin affect the efficiency of different types of glycotransferase and glycosidase on IgG, and whether the immobilization efficiency of IgG to protein A is affected by the reaction condition?

The reviewer is correct that IgG immobilization affects the efficiency of the glycoengineering reactions. We studied the conversion ratio of four glycosidase reactions with and without IgG immobilization (Figure S17). We found that immobilization leads to decreased efficiency, which tends to correlate with the size of glycoengineering enzymes. Such efficiency reduction likely results from the steric hindrance created by IgG-protein A interactions. Our computational modeling shows that binding to protein A decreases the accessibility of glycoengineering enzymes to the IgG Fc glycans. To address this issue, protein A engineering is worth to be investigated. For example, one can reduce the size of protein A (42 kD) by removing the motifs that are not involved in IgG binding. A paragraph of discussion about the immobilization steric effect is included in the supplementary material (Supplementary Discussion #4), as shown below: (reference citations included in the supplementary material)

*“IgG immobilization enables efficient washing and reaction-swapping processes in SPGR. However, we observed reduced enzyme activities on immobilized IgG compared to non-immobilized, free IgG (Figure S17a). Such a “trade-off” partially results from the relatively limited surface area in heterogeneous reactions but could mainly be attributed to the increased steric hindrance created by IgG binding to protein A. It’s reported that protein A binds to the IgG Fc at the interface between the CH2 and CH3 domains. This binding region is not only close to but also interacting with the Asn297 glycans. Computational modeling of interactions between the full-length protein A (with four domains) and the Fc region indicates steric hindrance, specifically at the CH2 region (Figure S17b-c). This could lead to reduced accessibility of the glycans by glycoengineering enzymes. This hypothesis of spatial hindrance and reduced accessibility is supported by the size effect of glycoengineering enzymes whose reduced activity correlates inversely with their molecular weight. For example, the largest enzyme in our toolset, Gal from *S. pneumoniae* (231kD), showed an activity reduction of 74% when functioning on immobilized IgG. A significant activity reduction was also found in the fucosidase reactions despite the smaller size of the enzyme (49kD) whose chemistry takes place at the reducing end of the glycan structure. In addition to steric hindrance, multiple IgG Fc regions could be interacting with the same protein A molecule, thus leading to a crowding effect and reduced enzymatic activity.”*

On the other hand, we found that the glycoengineering reaction does not affect IgG immobilization. Our newly added data (please see our responses to reviewer #1, and Figure S13) showed that the amount of IgG recovered from SPGR reactions is not dependent upon the enzyme species, reaction time, and buffer conditions. The use of high reaction temperature, however, decreased the product IgG recovery slightly. This might result from the dissociation of IgG from protein A resins during the reaction and/or the decreased protein stability at high temperatures.

2, Extensive screening works have been performed on the reaction conditions of different glycan-engineered enzymes, since the most appropriate reaction temperature and PH (5.5-7.5) value are different among various enzymes, how to ensure that the reaction conditions for the previous enzyme does not affect the reaction of the next one?.

A major advantage of running sequential reactions using the solid-phase platform is that the reaction buffer and enzymes can be easily replaced for the next reaction. Specifically, after each reaction, the buffer is removed from the IgG substrates (immobilized on resins) by either centrifuge or vacuum suction, followed by washing and re-conditioning with the new buffer. One can increase the amount of washing buffer used here to ensure that previous enzymes are removed completely.

To make this information clearer to the readers, we have revised the corresponding paragraph, as shown below:

“SPGR processes can be separated into 5 steps: 1) resin loading; 2) IgG immobilization; 3) washing and conditioning; 4) enzymatic glycan remodeling, and; 5) washing and elution. When running multi-step glycan remodeling, the third and fourth steps are repeated to swap the reaction buffer and enzymes. The substrate IgG remains immobilized on protein A resin until all the glycan remodeling steps are complete.”

3, For continuous reaction, the replacement of metal ion types is more cumbersome, is it possible to use one or two metal ions to simplify the glycosylation step?

The short answer is Yes. One can use a generic cation combination for all the glycosylation steps, such as 10 mM MgCl₂ and 10 mM MnCl₂. However, the activity of the enzymes could be compromised slightly in this case. The incubation time or enzyme concentration needs to be increased to ensure the complete conversion of the substrate to the product.

We would also like to point out that the use of the solid-phase platform has made the replacement of metal ions easy. In our standard protocols, as described in the supplementary material, we wash the resins with 20X bed volume washing buffer, followed by 10X bed volume of the new reaction buffer. We didn't observe the decrease of enzyme activity caused by leftover undesired cations.

REVIEWERS' COMMENTS:

Reviewer #2 (Remarks to the Author):

I think the revised manuscript is now in great shape to be published.

Reviewer #3 (Remarks to the Author):

The work by Mann et al is excellent and the logic of this manuscript is clear and well organized. To make SPGR approach more feasible and be used in the field of biopharmaceutical, I suggest detailed studies to check the effect of SPGR reaction conditions on the stability, structure and function of substrate antibody/protein and expect to read thorough results in following publication.